# Investigation of Synergistic Influence of Ultrasound and Co-Doping to Degrade Toluene from Polluted Air in Construction Sites—An Experimental Approach

Omid Akbarzadeh [1,2,3], Yahya Rasoulzadeh [1,4,*,†], Mohammad Haghighi [2,3,*,†], Azadeh Talati [2,3] and Hamed Golzad [5]

1. Department of Occupational Health Engineering, Faculty of Health, Tabriz University of Medical Sciences, Tabriz, P.O. Box 51656-65811, Iran; omid.akbrzdh@gmail.com
2. Chemical Engineering Faculty, Sahand University of Technology, Sahand New Town, Tabriz P.O. Box 51335-1996, Iran; azadeh_talati@yahoo.com
3. Reactor and Catalysis Research Centre (RCRC), Sahand University of Technology, Sahand New Town, Tabriz P.O.Box 51335-1996, Iran
4. Iranian Road Traffic Injury Research Centre, Tabriz University of Medical Sciences, Tabriz P.O. Box 51656-65811, Iran
5. School of Design and Built Environment, University of Canberra, Canberra 2617, Australia; hamed.golzad@canberra.edu.au
* Correspondence: rasoulzadehy@tbzmed.ac.ir (Y.R.); haghighi@sut.ac.ir (M.H.); Tel.: +98-41-33376228 (Y.R.); +98-41-33458096 or +98-41-33459152 (M.H.); Fax: +98-41-33376228(Y.R.); +98-41-33444355 (M.H.)
† These authors contributed equally to this work.

**Abstract: Background**: Toluene exposure in construction workers can lead to several health problems, primarily affecting the nervous system, respiratory system, and skin. Utilizing advanced photocatalytic materials to degrade gaseous toluene aims to significantly mitigate its negative impact. **Methods**: In this research, photocatalysts based on pure $TiO_2$ and modified $TiO_2$ were synthesized to evaluate their efficacy in degrading gaseous toluene, a prevalent air pollutant in construction settings. Two synthesis methods were employed. Sonoprecipitation was used to create Fe-N co-doped $TiO_2$ nanoparticles in the first method, while the second method utilized co-precipitation and hydrothermal techniques without ultrasonic assistance to achieve Fe-N co-doping. Seven types of nanophotocatalysts were synthesized, including $TiO_2$-U (with ultrasonic assistance), $NTiO_2$-U, $FeNTiO_2$ (2.5)-U, $FeNTiO_2$ (5)-U, $FeNTiO_2$ (7.5)-U, $FeNTiO_2$ (10)-U, and $FeNTiO_2$ (5) without ultrasonic assistance. Characterization of the synthesized photocatalysts involved various analyses, including XRD, SEM, EDX, UV–VIS DRS, FT–IR, BET, and $N_2$ adsorption-desorption isotherm. **Results**: Ultrasonic assistance notably improved particle dispersion and prevented agglomeration on the photocatalyst surface. UV–VIS DRS analysis indicated a reduction in band gap energy due to Fe and N doping of $TiO_2$. The study also investigated the influence of Fe doping, initial toluene concentration, light source, and residence time on the degradation rate of gaseous toluene. Experimental findings showed that $FeNTiO_2$ (5)-U exhibited a higher degradation rate of toluene (63.5%) compared to $FeNTiO_2$ (5) (50%) under visible light irradiation over 15 s. **Conclusions**: The study underscores the significant enhancement in photocatalytic activity for toluene degradation achieved through the combined effects of ultrasound and co-doping methods.

**Keywords:** $FeNTiO_2$; nanophotocatalyst; sonoprecipitation; construction workers; gaseous toluene

## 1. Introduction

Due to industrial growth, the presence of air pollutants, such as volatile organic compounds (VOCs) and polycyclic aromatic hydrocarbons (PAHs), has risen in both environmental and workplace settings. [1–3]. Petroleum refineries and storage facilities, residential heating systems, the use of solvents in construction and cleaning agents, as

well as tobacco smoke, constitute the primary sources of these pollutants [4–7]. Toluene, utilized in the production of various chemical compounds that are used widely in the construction process, such as paints, coatings, adhesives, sealants, waterproofing products, and polishing agents, represents a type of volatile organic compound (VOC) that poses risks to both humans and the environment [8–11]. Toluene vapors can be harmful when inhaled, leading to dizziness, headaches, and more severe neurological damage with prolonged exposure. Studies show that both acute and chronic exposure to toluene affects the central nervous system, leading to encephalopathy, cerebellar dysfunction, and other neurotoxic effects. Additionally, prolonged or repeated skin contact can cause irritation and dermatitis. The skin, being a route of exposure, absorbs toluene, which can lead to local and systemic toxic effects, emphasizing the importance of protective measures [12]. Furthermore, improper disposal and spillage can lead to soil and water contamination, posing risks to wildlife and ecosystems. Toluene's persistence in the environment and its toxicological profile make it a significant environmental pollutant [13].

Various techniques exist for removing pollutants, like toluene and benzene, from workplace air streams. Ideally, pollutants should be eliminated at source in the production process; when this is not feasible, workplace health and safety (WHS) management methods, such as engineering control, including ventilation, must be employed to degrade air pollutants. One method that has garnered significant interest among researchers is photocatalytic degradation. This process uses semiconductors, such as $TiO_2$-based photocatalysts, to break down air pollutants through photocatalysis [14–17]. $TiO_2$ is widely recognized as a high-performance photocatalyst extensively employed for degrading air pollutants, treating wastewater, and eliminating residual pesticides, among other applications [18–20]. $TiO_2$ will be active when exposed to UV light because it has a high-energy band gap (3.2 eV), so this is the main drawback of the $TiO_2$ photocatalyst. Therefore, to enhance photocatalyst efficiency in the visible light region, some metals, such as Fe, Co, Mn [21–23], and non-metals, such as N, S and C [24–26], can be doped into photocatalyst structures.

Introducing metal ions into $TiO_2$ traps electrons or holes, altering the recombination rate of electron–hole pairs [27]. Iron atoms have been regarded as suitable among various transition metals because the ionic radius of iron is closely matched with that of titanium. This similarity allows $Fe^{3+}$ ions to effectively integrate into the crystal lattice of $TiO_2$ [28]. Another limitation of $TiO_2$ photocatalysts is the rapid recombination of charge carriers, which occurs within nanoseconds in the absence of promoters [29]. Therefore, iron-doped $TiO_2$ mitigates the recombination of electron–hole pairs and enhances the photocatalytic activity [30,31]. Furthermore, researchers are increasingly focusing on the ultrasonic method for synthesizing photocatalysts [32–34]. This approach is beneficial for producing both amorphous and crystalline photocatalysts at the nanoscale, thereby enhancing photocatalytic activity in the visible light spectrum. [34,35]. Ultrasonic waves generate regions of high and low pressure, resulting in the collapse of cavities. This process creates localized hot spots of high temperature and pressure, facilitating the thorough mixing of elements at the nanoscale. Additionally, the formation and collapse of cavitation bubbles induce shock waves that break down particle aggregates and reduce particle size. [36–38].

Recent studies have demonstrated that doping $TiO_2$ with non-metal atoms from the P block of the periodic table shifts its optical absorption edge to lower energies. This enhancement is expected to improve photocatalytic efficiency in the visible light spectrum [39,40]. To put it differently, introducing oxygen vacancies by substituting lattice oxygen with elements from the P block, particularly through N-doping, reduces the band gap energy of $TiO_2$. This modification enhances the photocatalytic degradation efficiency under visible light conditions [41–44]. This has been explained as due to a similar atomic size to oxygen, small ionization energy and high stability [45].

The surface area, particle size, band gap energy, and crystalline structure of a photocatalyst, along with factors such as light intensity, pH, water vapor, and temperature, significantly influence the enhancement of photocatalytic degradation rates. Research highlights the critical role of the elemental composition in photocatalytic activity, underscoring

that synthesis conditions are crucial for determining catalyst properties. The method of synthesis directly affects the doping status of ions within the catalyst. [46–48]. Various synthesis methods, including sol–gel, hydrothermal, precipitation, and ultrasonic techniques, are available for producing photocatalysts. In catalyst synthesis, the sol–gel method offers high accuracy and uniformity but incurs higher costs and longer operation times, while the hydrothermal technique, though less expensive and faster, faces implementation difficulties due to stringent reaction conditions; precipitation is cost-effective and simple but may suffer from lower precision, and ultrasonic techniques provide rapid processing and enhanced dispersion at the expense of potential equipment-related challenges. In certain cases, combinations of two or three methods may be employed to synthesize a photocatalyst [49–52].

To address the shortcomings of the previous methods, this study aims to utilize ultrasound-assisted co-precipitation and hydrothermal methods to synthesize Fe-N co-doped $TiO_2$ photocatalysts and investigate the impact of ultrasound assistance on photocatalyst characterization and activity. In this study, the photocatalytic efficiency of both pure $TiO_2$ and modified $TiO_2$ are evaluated by their ability to degrade gaseous toluene under both ultraviolet and visible light irradiation conditions. Finally, the main contributions to the body of knowledge and practice can be listed as:

- The setting up of a continuous air purification reactor suitable for the real conditions of working environments;
- Increasing the contact surface of the pollutant with the photocatalyst and the retention time by creating sloping surfaces;
- Synthesis of a photocatalytic with better properties using an ultrasonic bath technique;
- Achieving a high rate of toluene degradation under continuous conditions;
- Improvement of photocatalyst performance under visible light compared to ultraviolet light.

## 2. Materials and Methods

### 2.1. Materials

$TiCl_4$ was employed as the titanium precursor for synthesizing $TiO_2$, HMT (hexamethylenetetramine) served as the nitrogen source, iron nitrate nonahydrate ($Fe(NO_3)_3 \cdot 9H_2O$) was used as the iron source, and a 25% ammonia solution was utilized for precipitation. All materials were either purchased from or produced by Merch Company (Los Angeles, CA, USA).

### 2.2. Nanophotocatalyst Preparation Procedure

As depicted in Figure 1, the samples were synthesized using two methods. In the first approach, Fe-N co-doped $TiO_2$ nanoparticles were prepared via sonoprecipitation. The second method involved synthesizing Fe-N co-doped $TiO_2$ through co-precipitation and hydrothermal techniques without ultrasonic assistance. In detail, a specific volume of $TiCl_4$, HMT, and $Fe(NO_3)_3 \cdot 9H_2O$ was added to 20 mL of deionized water. The mixture was stirred using a magnetic stirrer for 2.5 h at 40 °C. Subsequently, a 25% ammonia solution was added dropwise until the pH reached $8.0 \pm 2$. The suspension was vigorously stirred at 40 °C. The resulting stable solution underwent 45 min of sonication at 300 W (Figure 2). Another similar sample was prepared using conventional mixing with a magnetic stirrer for 45 min. After this step, hydrothermal aging was conducted for 12 h at 110 °C. Filtration and washing with deionized water were performed three times for all samples. The samples were then dried for 12 h at 110 °C in ambient air. Calcination was carried out for 5 h at 600 °C in ambient air.

Finally, nano-photocatalysts were synthesized in seven types: $TiO_2$-U (U denotes ultrasonic-assisted), $NTiO_2$-U, $FeNTiO_2$ (2.5)-U, $FeNTiO_2$ (5)-U, $FeNTiO_2$ (7.5)-U, $FeNTiO_2$ (10)-U, and $FeNTiO_2$ (5), without ultrasonic assistance.

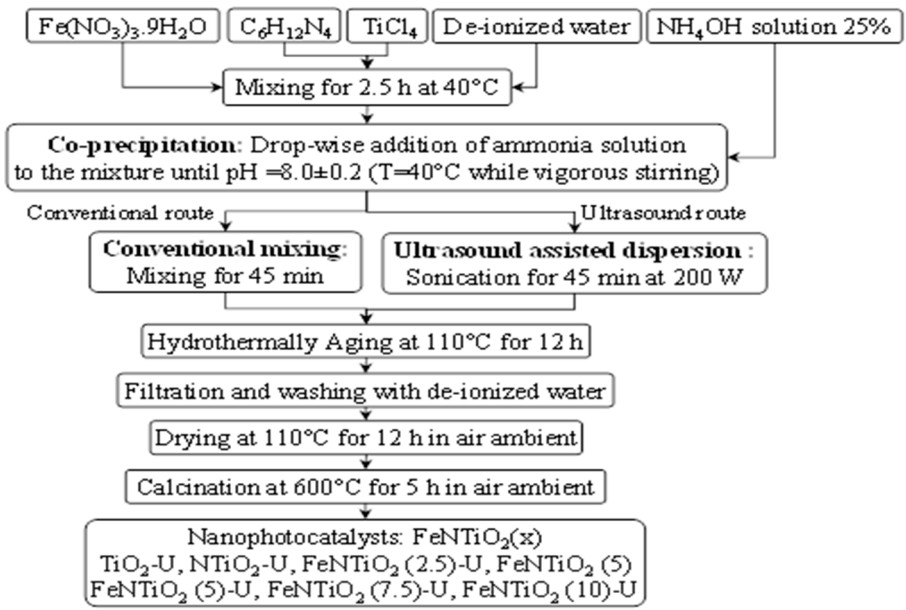

**Figure 1.** One-pot sonoprecipitation design of FeNTiO$_2$ nanophotocatalyst via ultrasound and co-doping synergistic method.

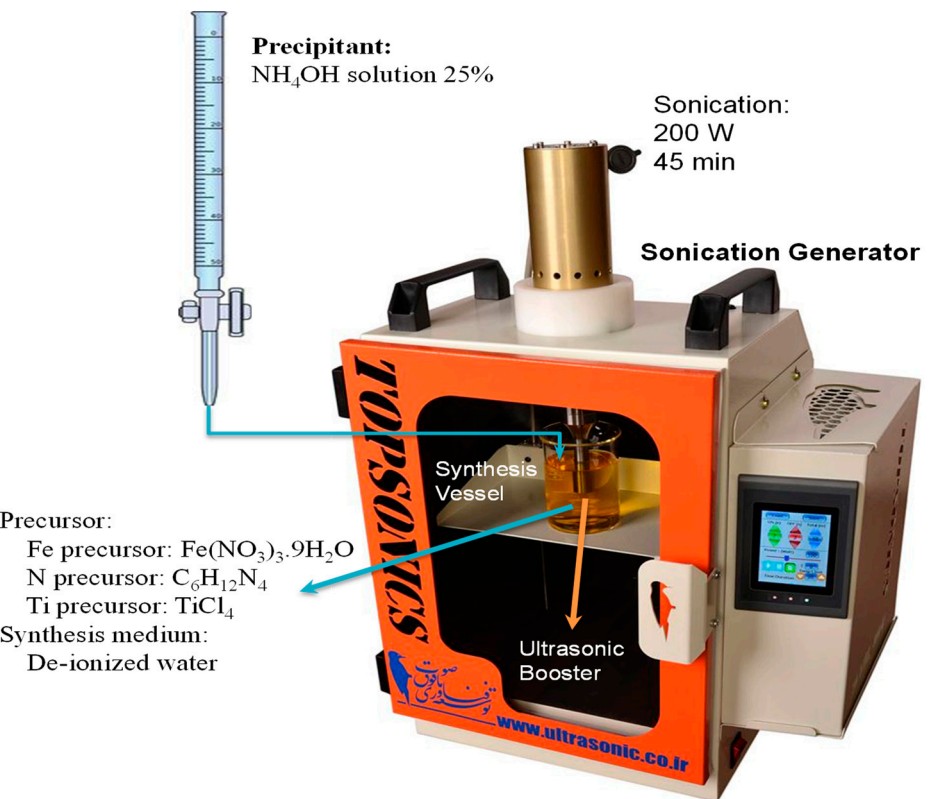

**Figure 2.** Experimental setup for one-pot sonoprecipitation design of FeNTiO$_2$ nanophotocatalyst via ultrasound and co-doping synergistic method.

### 2.3. Nanophotocatalysts Characterization Techniques

To examine the crystalline structure of the samples, X-ray diffraction was performed over a 2θ range of 5–70 degrees at a temperature of 25 °C using a Panalytical Xpert PRO X-ray Diffractometer. BET surface analysis was conducted to determine the specific surface area through nitrogen adsorption at 77 K using a Specific Surface Area and Porosity Analyzer, PHS 1020.

The morphology of the synthesized photocatalyst was examined using a scanning electron microscope (Philips XL30 ESEM). Fourier transform infrared spectra (FTIR) were analyzed using an FT–IR/FT–NIR Spectrum (400 Spectrometer). UV–VIS diffuse reflectance spectroscopy (UV-VIS DRS) was performed using a Shimadzu UV 160 A UV–Vis Spectrometer to study the optical properties.

Energy Dispersive X-ray spectroscopy (EDS) was utilized with an EDX spectrometer integrated with the SEM to analyze the elemental composition of the photocatalysts.

### 2.4. Experimental Setup for Photocatalytic Performance Test

To assess the photocatalytic performance of the synthesized photocatalysts, experiments were conducted under uniform conditions using $TiO_2$-U, N-$TiO_2$-U, Fe-N/$TiO_2$ (5)-U, and Fe-N/$TiO_2$ (5). Figure 3 illustrates the experimental configuration. The photocatalytic reactions took place in a 180 $cm^3$ two-stage reactor constructed from polytetrafluoroethylene (PTFE). Two quartz glass panels, each 1 mm thick, were fitted onto the PTFE. To enhance air turbulence within the reactor and improve the interaction between the gas flow and the photocatalyst, the PTFE surface was corrugated using a CNC machine (CNC NRC 6090 S2). Toluene was prepared in four concentration ranges: 50 ppm, 100 ppm, 150 ppm, and 200 ppm, within a chamber measuring 50 cm × 50 cm × 20 cm, while airflow was regulated by two rotameters. A thin layer of concrete (sand mesh: 20 × 40) was applied on the PTFE, with the photocatalyst then coated on this layer. At the conclusion of each experiment, the concrete layer was removed, and a new one was created on the PTFE. Two UVA Philips 60 W lamps were positioned 1 cm away from the glass panels, and additional LED lamps were utilized for visible light photocatalytic degradation. Air containing toluene was circulated through the reactor, and the outlet concentration was measured 15 s later using a direct reading first check (first check + multi-gas PID, Ion technology). The degradation rate was determined by calculating the ratio of the outlet concentration to the inlet concentration.

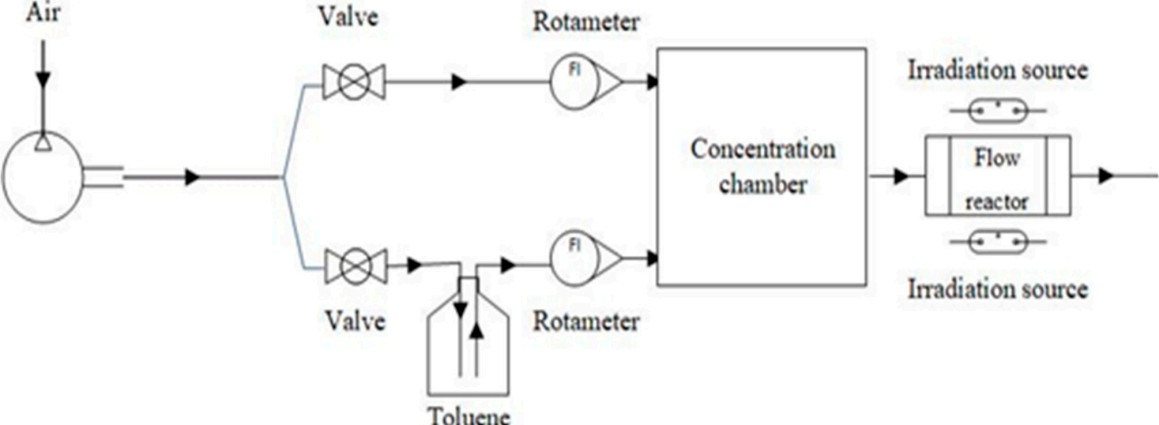

**Figure 3.** Experimental setup for testing of photocatalytic performance of co-doped FeN$TiO_2$ nanophotocatalyst in efficient visible light responsive photodecomposition of toluene from polluted air in a flow reactor.

## 3. Results and Discussions

### 3.1. Nanophotocatalysts Characterization

#### 3.1.1. XRD Analysis

Figure 4 displays the XRD patterns of $TiO_2$-U, N-$TiO_2$-U, Fe-N/$TiO_2$ (5)-U, and Fe-N/$TiO_2$ (5). The peaks corresponding to $TiO_2$-U and the other samples appear at 2θ = 25.3, 37.9, 48.4, 53.9, 55.3, 62.7, 69.0, 70.2, 75.4, and 83.2 (see Table 1), which aligns with the JCPDS: 00-001-0562 for anatase $TiO_2$, indicating the successful synthesis of the photocatalysts. Notably, only the anatase phase is present in all samples, with no evidence of rutile or brookite phases. Upon the addition of iron and nitrogen, the XRD peaks become broader

and shorter, suggesting the incorporation of these dopants into the $TiO_2$ structure and a reduction in crystallite size. Despite doping with N and Fe in various ratios, the anatase phase of $TiO_2$ remains unchanged. The crystallinity of $TiO_2$, however, diminishes with the inclusion of nitrogen and iron (see Table 2). The absence of distinct peaks for iron and nitrogen in the patterns is attributed to the similarity in ionic radii between $Fe^{3+}$ and $Ti^{4+}$, resulting in the integration of $Fe^{3+}$ ions into the $TiO_2$ lattice. Similar XRD pattern results were reported by Kalantari et al. [53]. Additionally, a comparison between the non-ultrasound and ultrasound patterns of Fe-N/$TiO_2$ (5) reveals that the Fe-N/$TiO_2$ (5)-U sample has a superior crystalline structure. This improvement is attributed to the use of the sonochemistry method, which enhances the distribution of crystal size and structure [54].

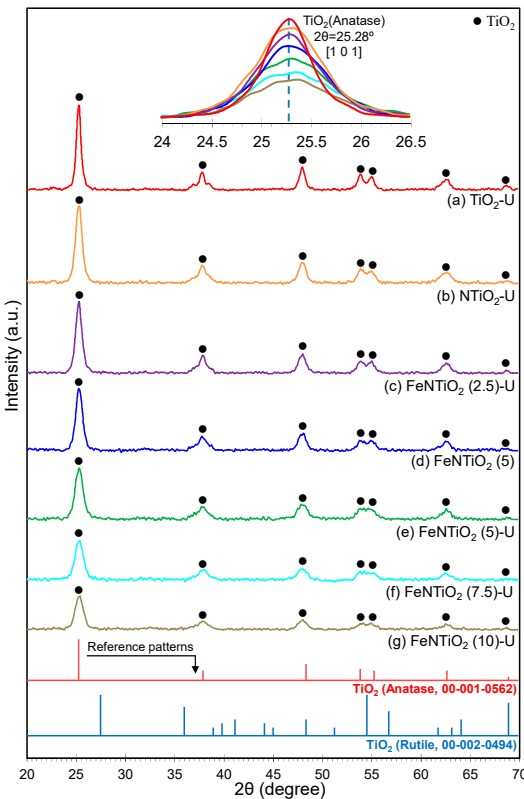

**Figure 4.** XRD patterns of co-doped FeNTiO$_2$ nanophotocatalysts: (**a**) TiO$_2$-U, (**b**) NTiO$_2$-U, (**c**) FeNTiO$_2$ (2.5)-U, (**d**) FeNTiO$_2$ (5), (**e**) FeNTiO$_2$ (5)-U, (**f**) FeNTiO$_2$ (7.5)-U and (**g**) FeNTiO$_2$ (10)-U.

**Table 1.** XRD Major Peaks of TiO$_2$.

| No | Substance | JCPDS No | Phase | XRD Major Peaks |
|----|-----------|----------|-------|-----------------|
| 1 | TiO$_2$ | 00-001-0562 | Anatase | 25.3, 37.9, 48.4, 53.9, 55.3, 62.7, 69.0, 70.2, 75.4, 83.2 |

**Table 2.** Structural properties of co-doped FeNTiO$_2$ nanophotocatalyst.

| Nanophotocatalyst | N (wt. %) | Fe/TiO$_2$ (wt. %) | Ultrasound Irradiation | | S$_{BET}$ (m$^2$/g) | V$_P$ (cm$^3$/g) | D$_P$ (nm) | λ$_0$ (nm) | Band Gap (eV) | Relative Crystallinity TiO$_2$ [a] |
|---|---|---|---|---|---|---|---|---|---|---|
| | | | Time (min) | Power (W) | | | | | | |
| TiO$_2$-U | - | 0 | 45 | 300 | 47.8 | 0.1310 | 10.6 | 390.5 | 3.18 | 100 |
| NTiO$_2$-U | 30 | 0 | 45 | 300 | 88.0 | 0.1880 | 8.2 | 413.3 | 3 | 92.6 |
| FeNTiO$_2$(2.5)-U | 30 | 2.5 | 45 | 300 | - | - | - | - | - | 86.2 |
| FeNTiO$_2$(5) | 30 | 5 | - | - | 89.4 | 0.1940 | 8.1 | 424.7 | 2.92 | 75.5 |
| FeNTiO$_2$(5)-U | 30 | 5 | 45 | 300 | 88.3 | 0.2070 | 9.3 | 437.7 | 2.83 | 61.7 |
| FeNTiO$_2$(7.5)-U | 30 | 7.5 | 45 | 300 | - | - | - | - | - | 48.9 |
| FeNTiO$_2$(10)-U | 30 | 10 | 45 | 300 | - | - | - | - | - | 40.4 |

[a] Crystallite phase: Anatase (JCPDS: 00-001-0562, 2θ = 25.3, 37.9, 48.4, 53.9, 55.3, 62.7, 69.0, 70.2, 75.4 and 83.2°).

3.1.2. FESEM Analysis

Figure 5 presents the field emission scanning electron microscopy (FESEM) images of TiO$_2$-U, N-TiO$_2$-U, Fe-N/TiO$_2$ (5)-U, and Fe-N/TiO$_2$ (5) photocatalysts. The analysis reveals that all samples are nanoscale with a spherical shape. Specifically, TiO$_2$-U particles are spherical and tend to agglomerate. The incorporation of iron and nitrogen into the TiO$_2$ structure reduces both particle size and agglomeration. Figure 5c,d illustrates the surfaces of Fe-N/TiO$_2$ (5) and Fe-N/TiO$_2$ (5)-U, respectively. It is evident that Fe-N/TiO$_2$ (5)-U particles are smaller and show no signs of agglomeration, indicating that the ultrasonic method enhances the distribution of Fe and N on the TiO$_2$ surface. The average particle sizes for Fe-N/TiO$_2$ (5)-U and Fe-N/TiO$_2$ (5) are 37.7 nm and 41.8 nm, respectively.

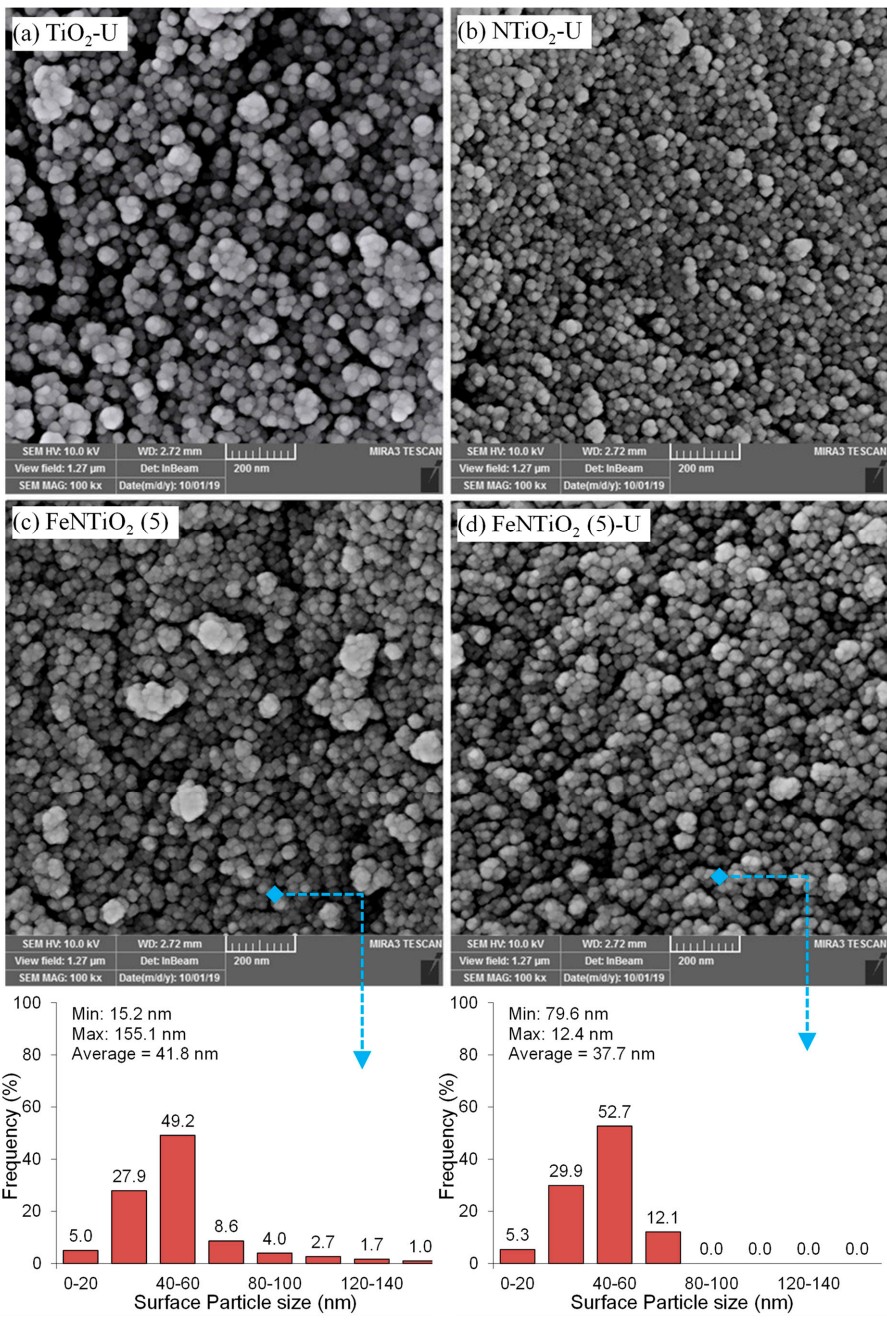

**Figure 5.** FESEM images of co-doped FeNTiO$_2$ nanophotocatalysts: (**a**) TiO$_2$-U, (**b**) NTiO$_2$-U, (**c**) FeNTiO$_2$ (5) and (**d**) FeNTiO$_2$ (5)-U.

Figure 6 depicts the 3D surface images of Fe-N/TiO$_2$ (5) (Figure 6b) and Fe-N/TiO$_2$ (5)-U (Figure 6a). The maximum height, average height, and root mean square for Fe-N/TiO$_2$ (5) and Fe-N/TiO$_2$ (5)-U are 200 nm, 91.9 nm, 33.9 nm, 50 nm, 25.5 nm, and 7.2 nm, respectively. These findings confirm that the use of the ultrasonic method results in reduced particle size and decreased agglomeration.

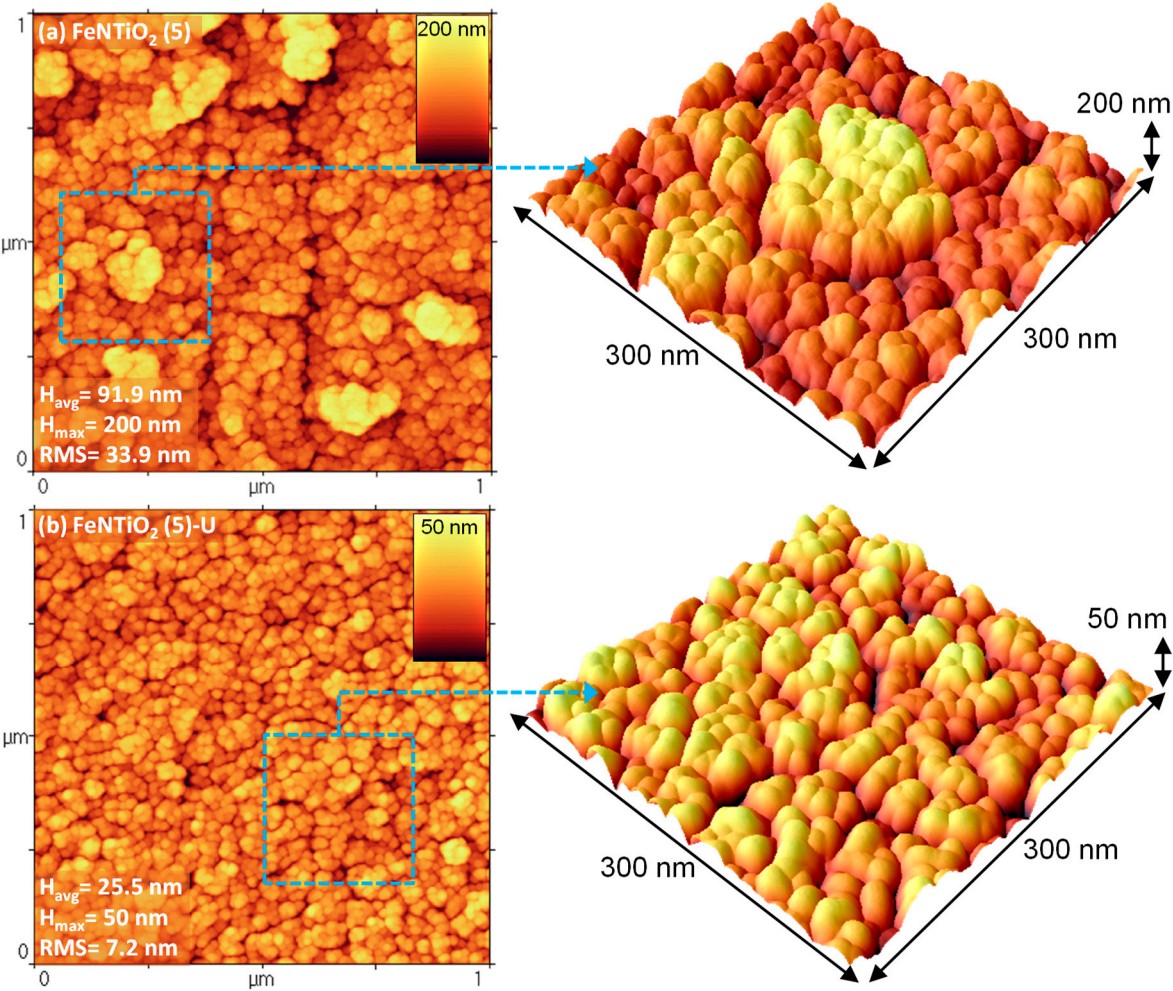

**Figure 6.** Overview of 3D surface images of co-doped FeNTiO$_2$ nanophotocatalysts: (**a**) FeNTiO$_2$ (5) and (**b**) FeNTiO$_2$ (5)-U.

### 3.1.3. EDX Analysis

Energy Dispersive Spectroscopy (EDS) was conducted alongside the SEM analysis using the same device under vacuum conditions to examine the elemental composition of pure TiO$_2$ and modified TiO$_2$. Figure 7 displays the EDX dot maps for TiO$_2$-U, N-TiO$_2$-U, Fe-N/TiO$_2$ 5-U, and Fe-N/TiO$_2$ 5. The EDX dot mapping confirms the presence of iron (Fe), nitrogen (N), titanium (Ti), and oxygen (O) in the samples. These results demonstrate that doping TiO$_2$ with Fe and N was successful, even though the XRD patterns did not show any peaks for Fe and N in the anatase TiO$_2$. Similar findings have been reported by Ganesh et al. and Ambati et al. [55,56]. Additionally, the EDX dot map micrographs reveal an enhanced distribution of Fe and N on the surface of TiO$_2$, with no evidence of agglomeration. The distribution of Fe-N/TiO$_2$ (5)-U is significantly better than that of the non-ultrasonic Fe-N/TiO$_2$ (5) sample. Agglomeration is observed on the surface of Fe-N/TiO$_2$ (5), but not on Fe-N/TiO$_2$ (5)-U. These findings suggest that the ultrasonic method improves the distribution of dopants on the TiO$_2$ surface, thereby enhancing the

photocatalytic activity of the samples. Zarrabi et al. [57] obtained similar results regarding the impact of the ultrasonic method on particle distribution.

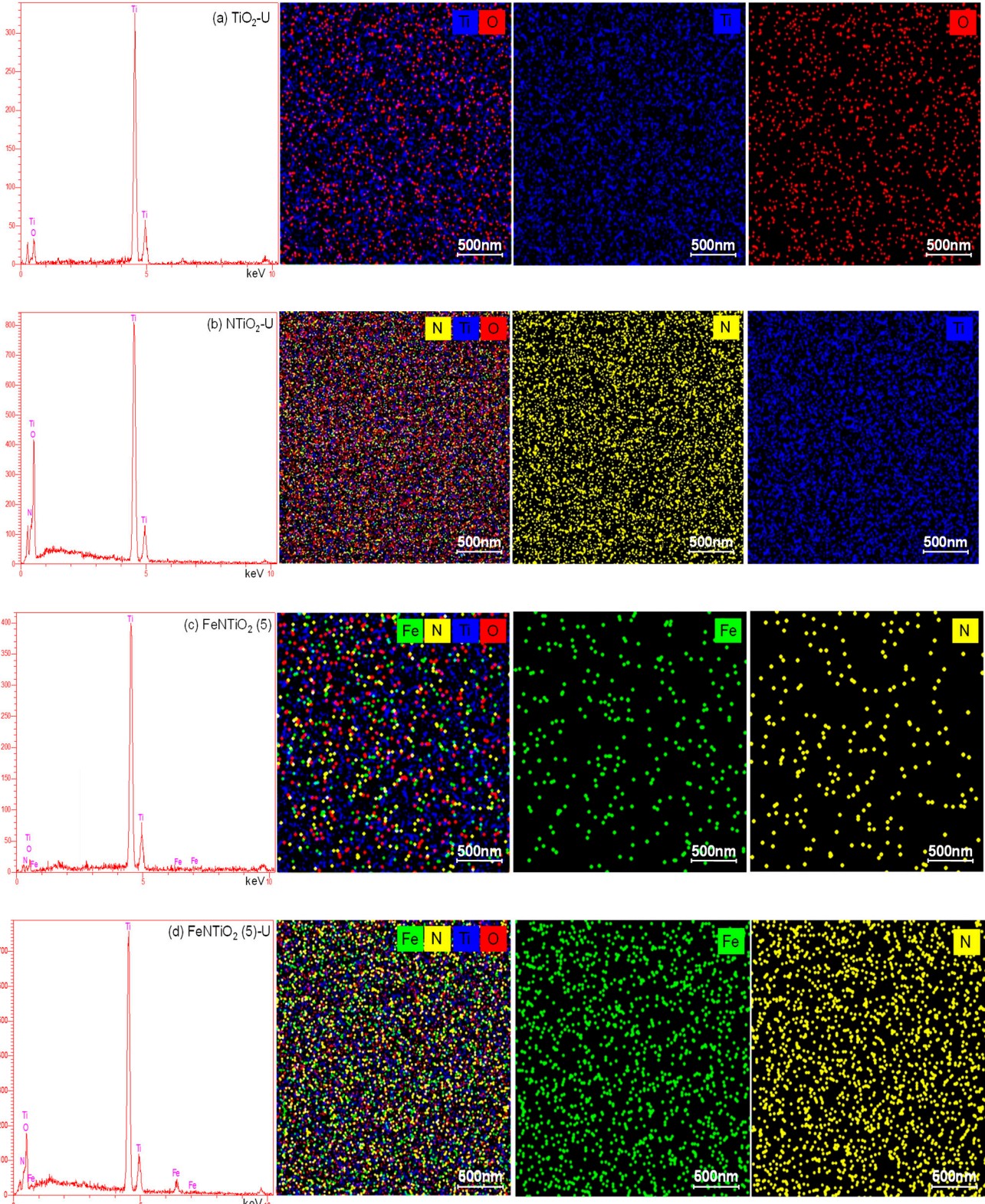

**Figure 7.** EDX analysis of co-doped $FeNTiO_2$ nanophotocatalysts: (**a**) $TiO_2$-U, (**b**) $NTiO_2$-U, (**c**) $FeNTiO_2$ (5) and (**d**) $FeNTiO_2$ (5)-U.

### 3.1.4. BET–BJH Analysis

The specific surface area ($S_{BET}$) plays a crucial role in photocatalytic activity. Figure 8 presents the BET–BJH analysis, adsorption and desorption graphs, and pore volume of the synthesized photocatalysts. The analysis indicates that the specific surface areas of modified $TiO_2$ are greater than that of pure $TiO_2$. Additionally, co-doped samples exhibit larger specific surface areas compared to mono-doped samples. Consequently, the particle size of co-doped samples is smaller than that of mono-doped and pure $TiO_2$. The sonication method helps prevent particle agglomeration and induces cavitation, which enhances the specific surface areas of both co-doped and mono-doped $TiO_2$ [58]. According to the results, the specific surface area of Fe-N/$TiO_2$ (5) U (88/37 $m^2 \cdot g^{-1}$) is less than Fe-N/$TiO_2$ (5) (89.4 $m^2 \cdot g^{-1}$), which could be due to measurement errors in the laboratory instruments. However, Asl et al. [59] reported similar results for specific surface area.

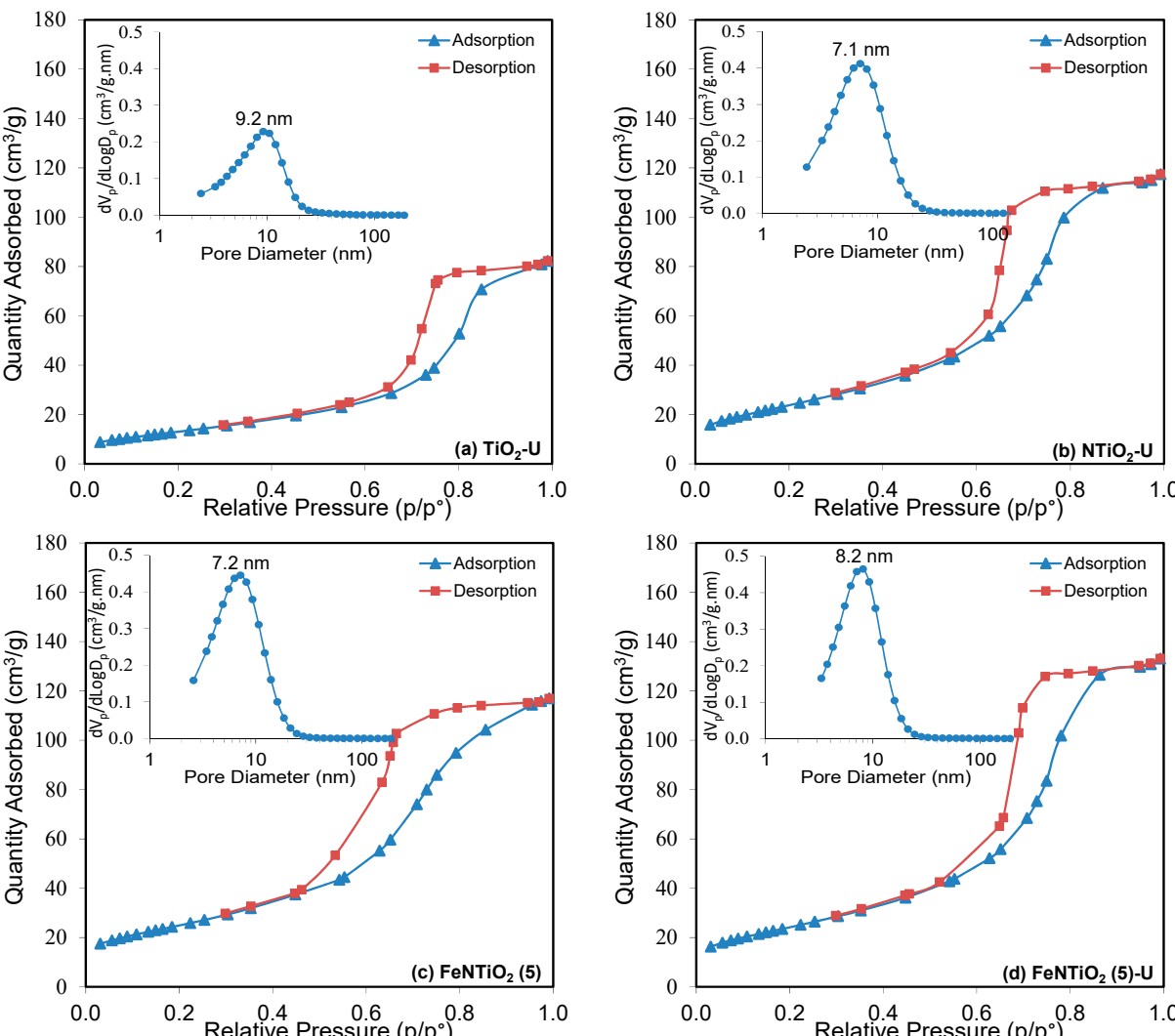

**Figure 8.** Adsorption/Desorption isotherms and pore size distribution of co-doped FeNTiO$_2$ nanophotocatalysts: (**a**) TiO$_2$-U, (**b**) NTiO$_2$-U, (**c**) FeNTiO$_2$ (5) and (**d**) FeNTiO$_2$ (5)-U.

Additionally, Figure 8 displays the $N_2$ adsorption–desorption isotherms, pore volume, and pore size distribution for $TiO_2$ and modified $TiO_2$. All samples exhibit a pore volume greater than 2 nm, indicating that $TiO_2$ has a mesoporous structure. The incorporation of Fe and N into $TiO_2$ results in an increase in the photocatalyst's pore volume. Initially, at low relative pressures, the adsorption of samples rises with increasing relative pressure. However, once the relative pressure reaches 0.6, the adsorption rate increases sharply due

to the accumulation of $N_2$ in the $TiO_2$ mesopores [60]. The samples exhibit the following order of adsorption capacity from highest to lowest: Fe-N/$TiO_2$ (5)-U > Fe-N/$TiO_2$ (5) > N/$TiO_2$-U > $TiO_2$-U. Notably, Fe-N/$TiO_2$ (5)-U shows a higher adsorption capacity compared to the non-ultrasonic synthesized Fe-N/$TiO_2$ (5), attributed to its larger and more diverse pore size distribution. The adsorption–desorption isotherms for these samples fall into the IV type and H1 hysteresis category. These findings indicate that Fe-N/$TiO_2$ (5)-U demonstrates superior photocatalytic activity compared to Fe-N/$TiO_2$ (5) and the other samples [61]. Moreover, the results show that employing the sonochemistry method enhances the photocatalytic properties of the samples, leading to more effective removal of pollutants from air and wastewater.

### 3.1.5. FTIR Analysis

Figure 9 displays the FTIR spectra for both $TiO_2$ and modified $TiO_2$. The broad and intense peaks observed in the range of 480–1050 cm$^{-1}$ are indicative of Ti-O and Ti-O-Ti bonds within the photocatalysts [62]. The peaks at 1450, 1620, and 3420 cm$^{-1}$ correspond to the presence of water on the photocatalyst surface, which is either physically adsorbed or associated with OH groups [63,64]. Additionally, the peak at 3420 cm$^{-1}$ is associated with the bonded OH groups, while the peak at 1620 cm$^{-1}$ corresponds to the O-H bending of water adsorbed on $TiO_2$ [65,66].

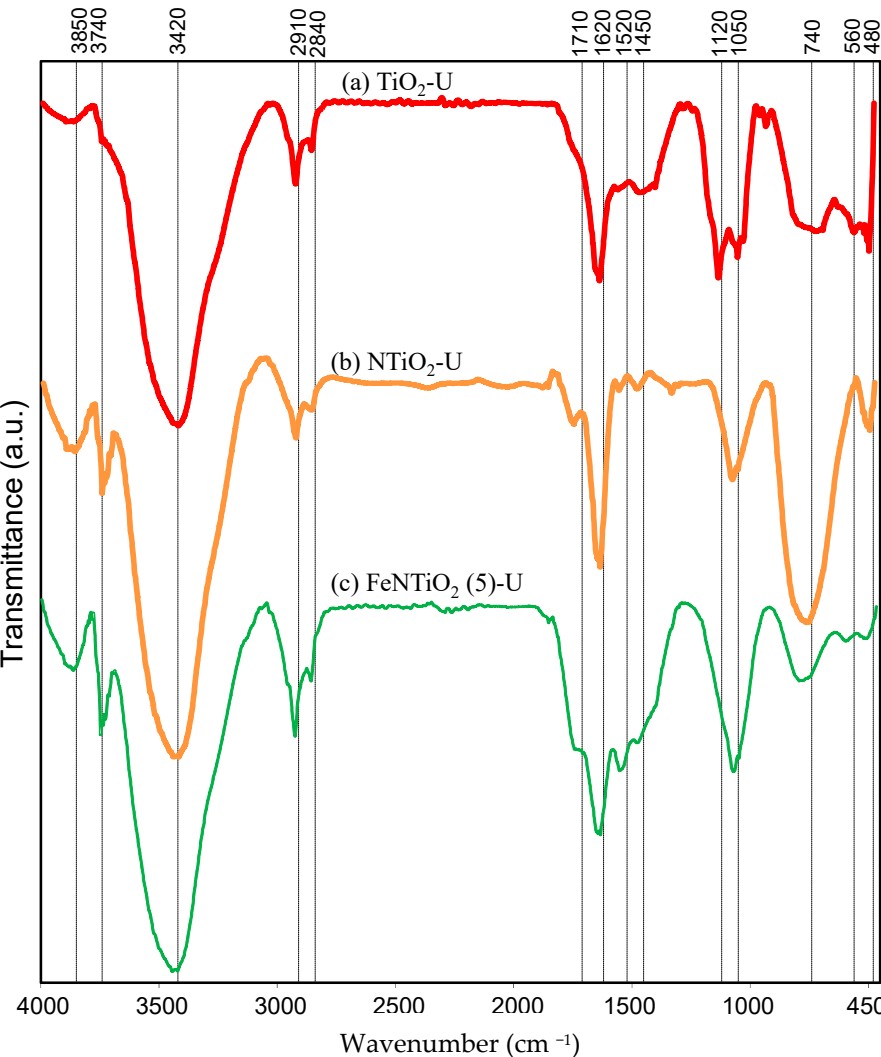

**Figure 9.** FTIR analysis of co-doped FeNTiO$_2$ nanophotocatalysts: (**a**) TiO$_2$-U, (**b**) NTiO$_2$-U and (**c**) FeNTiO$_2$ (5)-U.

Furthermore, the peaks at 2840 and 2910 $cm^{-1}$ are attributed to the vibrations of atmospheric C-H or $CO_2$ present on the photocatalyst surface. Additionally, a small peak observed in the 480–520 $cm^{-1}$ range is linked to metal oxides, specifically the IR vibrations of the anatase phase of Ti-O. In N-doped samples, this 480–520 $cm^{-1}$ peak is extended, indicating the formation of non-metal oxides [55,67]. Lastly, the FTIR analysis results show a significant correlation with the findings from EDX and XRD analyses.

### 3.1.6. DRS Analysis

Figure 10 presents the UV–VIS diffuse reflectance spectra for $TiO_2$ and modified $TiO_2$ photocatalysts. Among all the samples, $TiO_2$-U exhibits the lowest absorption, while $FeNTiO_2$ (5)-U shows the highest absorption. The band gap energies were determined using the formula $h\upsilon = 1240/\lambda_0$, where $h$ is Planck's constant, $\upsilon$ is frequency, and $\lambda_0$ is the maximum wavelength. The maximum wavelength gradients for $TiO_2$-U, $FeTiO_2$ (2.5)-U, $FeNTiO_2$ (5), and $FeNTiO_2$ (5)-U are 390.5, 413.3, 424.7, and 437.7 nm, respectively. Consequently, the band gap energies were calculated as 3.18, 3.00, 2.92, and 2.83 eV, respectively. The results indicate that doping $TiO_2$ with Fe and N shifts the light absorption edge and reduces band gap energies. Additionally, using the ultrasonic method in synthesizing $FeNTiO_2$ (5)-U increases absorption and decreases band gap energy, demonstrating that this method enhances light absorption by reducing particle size and increasing the number of active sites on the photocatalyst surface [68].

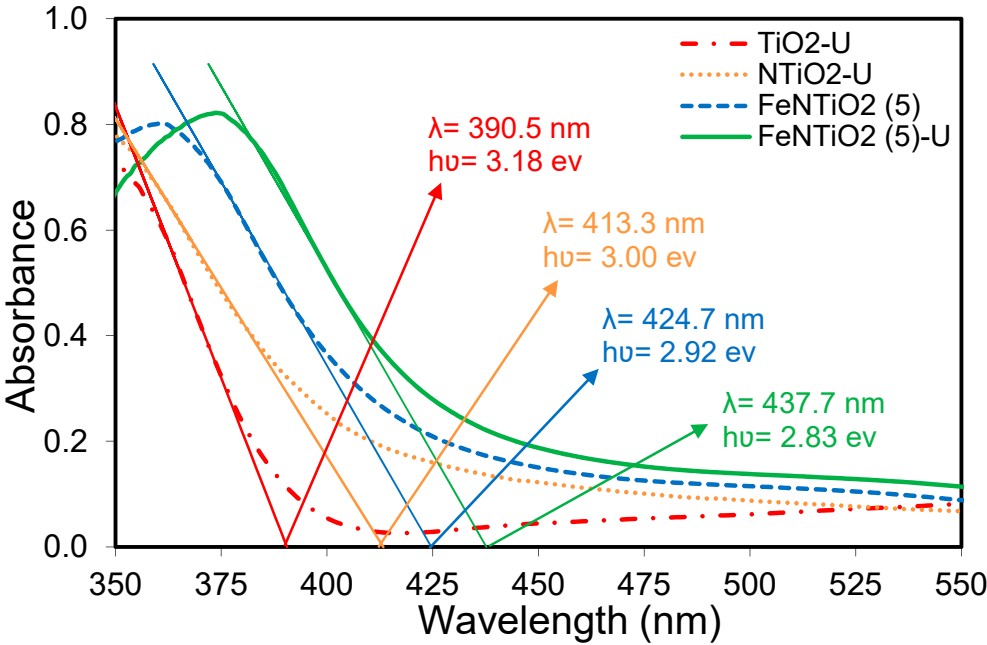

**Figure 10.** DRS analysis of co-doped $FeNTiO_2$ nanophotocatalysts: $TiO_2$-U, $NTiO_2$-U and $FeNTiO_2$ (5)-U.

Furthermore, Fe ions introduce a new energy level below the conduction band of $TiO_2$. Additionally, N doping in the $TiO_2$ structure lowers the band gap energies and enhances visible light absorption. This is due to the mixing of the N 2p energy state with the O 2p level at the top of the valence band of $TiO_2$ [69,70]. Moreover, it is evident that the band gap energies and visible light absorption of Fe and N co-doped $TiO_2$ are greater than those of N-doped $TiO_2$ and pure $TiO_2$. This is attributed to the charge transfer transition between the d-electrons of iron ions and the conduction or valence band of $TiO_2$ [71–73]. Additionally, iron ions can act as traps for photo-induced electrons. Due to their ability to reduce photo-induced electrons, Fe ions can capture holes at $Fe^{2+}$ sites, leading to the formation of $Fe^{3+}$ ions [74].

### 3.2. Photocatalytic Degradation of Toluene

### 3.2.1. Influence of Doping Composition

Figure 11 illustrates the photocatalytic degradation rate of toluene using $TiO_2$ U, $N/TiO_2$ U, $Fe-N/TiO_2$ (5) U, and $Fe-N/TiO_2$ (5) photocatalysts. At a concentration of 50 ppm, the toluene conversion after 15 s reached 22.40%, 23.30%, 40.2%, and 36.1%, respectively. These results indicate that doping $TiO_2$ with Fe and N enhances the photocatalytic degradation of toluene. The degradation rate is further improved when Fe is doped into the samples using the ultrasound-assisted method. XRD, SEM, and BET analyses confirm that doping with Fe and N reduces crystalline size and increases surface area, while also decreasing agglomeration. Additionally, there is only a slight increase in the degradation rate with N doping alone, which may be due to the removal and washing away of nitrogen during the synthesis process.

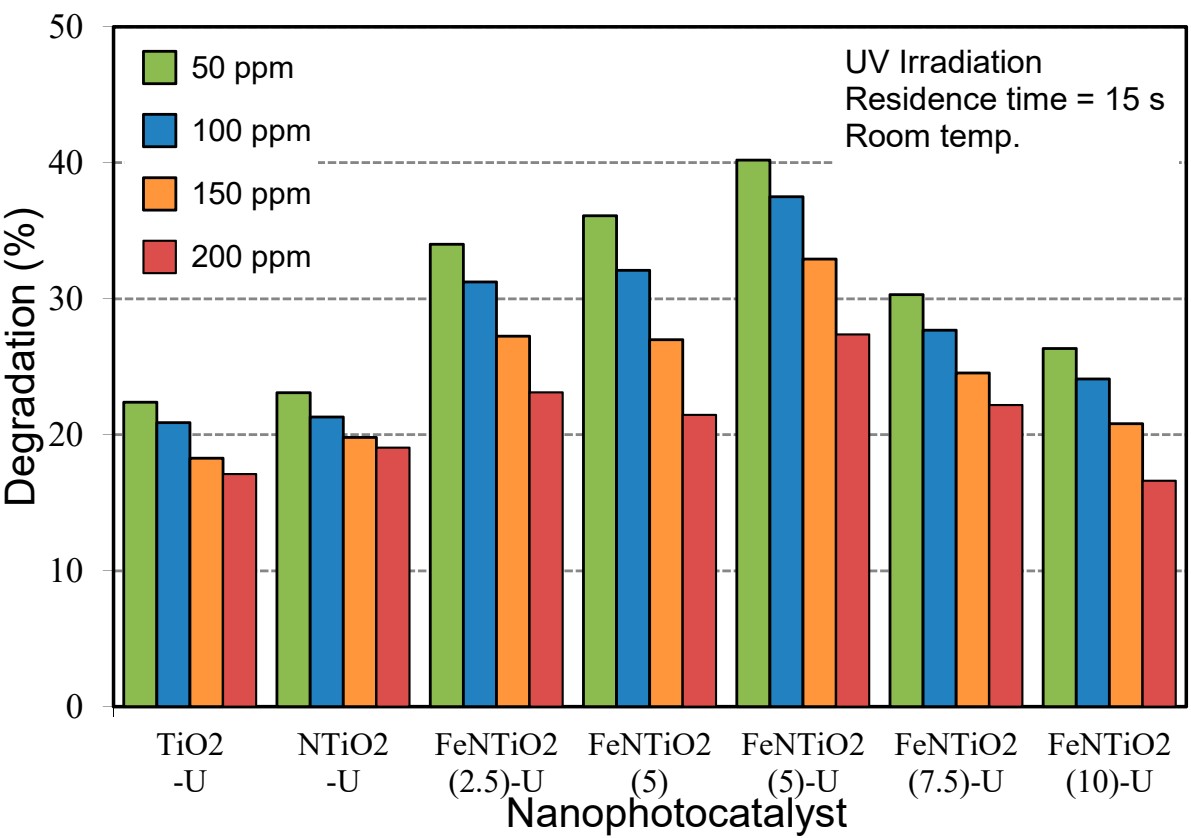

**Figure 11.** Influence of doping composition on photocatalytic degradation of various toluene concentrations over $FeNTiO_2$ nanophotocatalysts under UV irradiation.

### 3.2.2. Influence of Ultrasound Irradiation

Figure 12 illustrates the impact of ultrasound-assisted versus conventional methods on the degradation of gaseous toluene. There is a clear distinction in photocatalytic activity between $FeNTiO_2$ (5)-U and $FeNTiO_2$ (5). The sample synthesized using the sonochemical method exhibits significantly higher photocatalytic activity compared to the sample made using the conventional method. This enhanced activity is attributed to the increased number of active sites, improved nucleation, and reduced particle size in the ultrasonic-assisted sample compared to the conventional one [75]. These results demonstrate that the ultrasonic method can enhance photocatalytic activity, effectively improving the removal of pollutants from the air. [76].

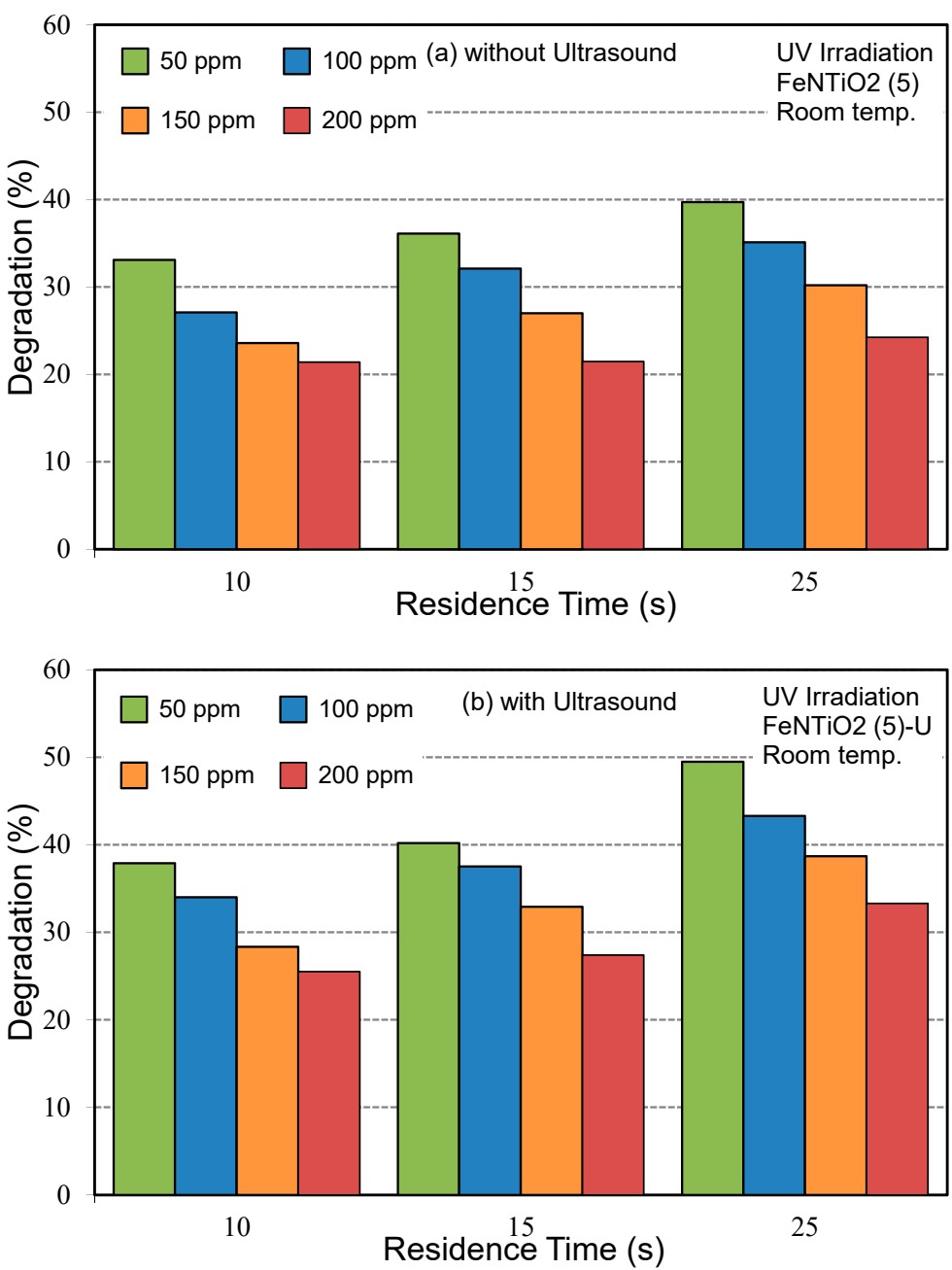

**Figure 12.** Influence of ultrasound irradiation at various residence times on co-doping of FeNTiO$_2$ nanophotocatalysts in photocatalytic degradation of gaseous toluene under UV irradiation: (**a**) without ultrasound and (**b**) with ultrasound.

Additionally, the impact of varying residence time was examined for Fe-N/TiO$_2$ (5) U and Fe-N/TiO$_2$ (5) samples (see Figure 12). As residence time increased and airflow velocity decreased, the degradation rate of the samples improved. This effect is more pronounced in the ultrasonic-assisted sample. This can be attributed to the extended residence time allowing for longer adsorption, which enhances the contact between photocatalysts and toluene molecules, thereby facilitating more effective oxidation of toluene [77]. Additionally, these results indicate that the degradation of toluene is influenced by the mass transfer to the photocatalyst surface. As residence time decreases and airflow rate increases, the photocatalytic degradation of gaseous toluene diminishes. This reduction is due to the shorter residence time, which limits the amount of toluene that can reach the photocatalyst

surface [78]. A study by Korologos et al. [79] reported comparable findings regarding residence time.

### 3.2.3. Influence of Light Source

The photocatalytic activity of both pure and modified $TiO_2$ was evaluated through the removal of gaseous toluene under UV and visible light to assess the performance of the samples and the effect of different light sources. Initially, experiments conducted without UV or visible light showed no significant reduction in toluene concentration after one reaction cycle, indicating that toluene remains stable in the presence of photocatalysts without light. Figure 13 illustrates the impact of light sources on toluene degradation rates. The degradation rate significantly increased when switching from UV irradiation to visible light (from 40.2% to 63.5% for Fe-N/$TiO_2$ (5) U. UV–VIS DRS analysis shows that doping $TiO_2$ with Fe and N enhances adsorption in the visible light range and reduces the band gap energy. These factors contribute to the improved photocatalytic activity of Fe and N co-doped $TiO_2$ under visible light, surpassing its performance under UV irradiation. Similar findings were reported by Dolat et al. [80].

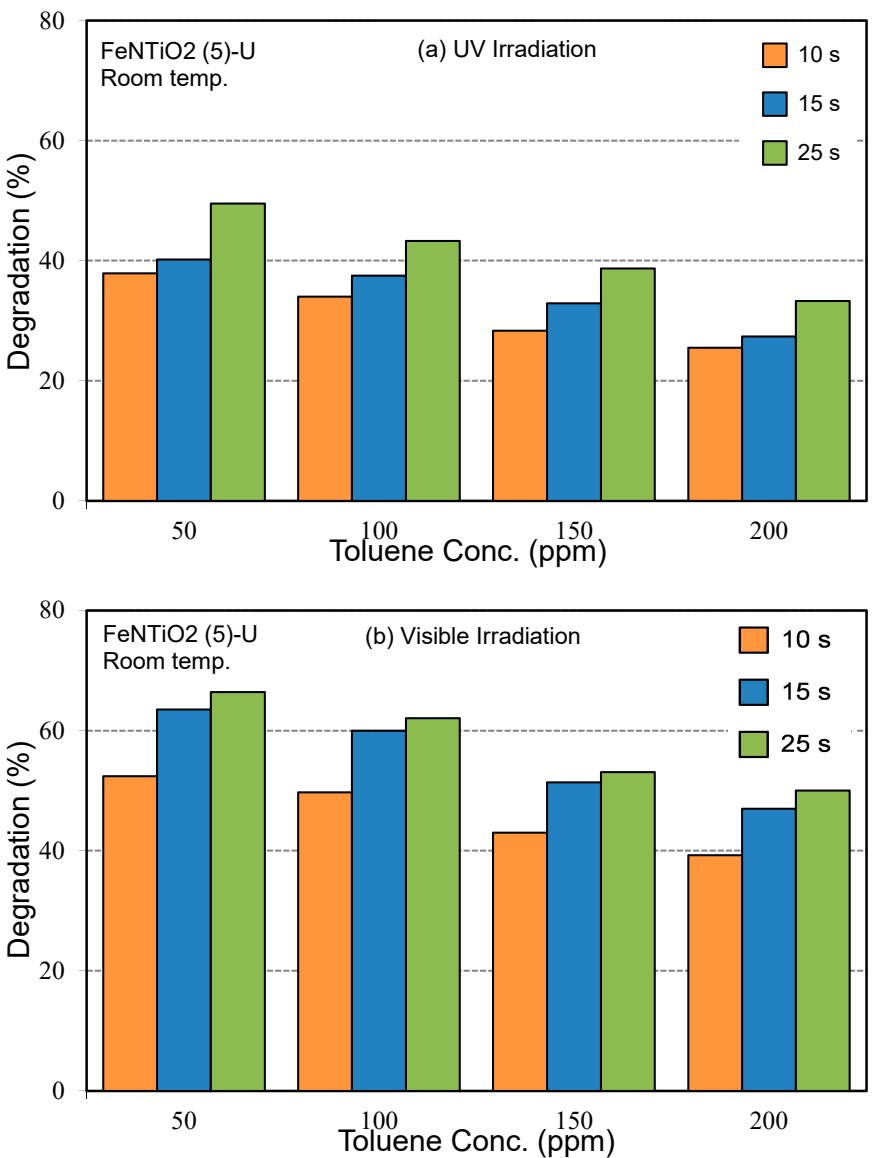

**Figure 13.** Influence of light source on photocatalytic degradation of various toluene concentrations over FeNTiO$_2$ (5)-U nanophotocatalyst.

### 3.3. Reaction Pathway for Toluene Photocatalytic Degradation

To study the mechanism of the photocatalytic degradation process for removing gaseous toluene (Figure 14), the energy value of the valance and conduction band was calculated by:

$$E_{vb} = X - E_0 + 0.5E_g$$

$$E_{cb} = E_{vb} - E_g$$

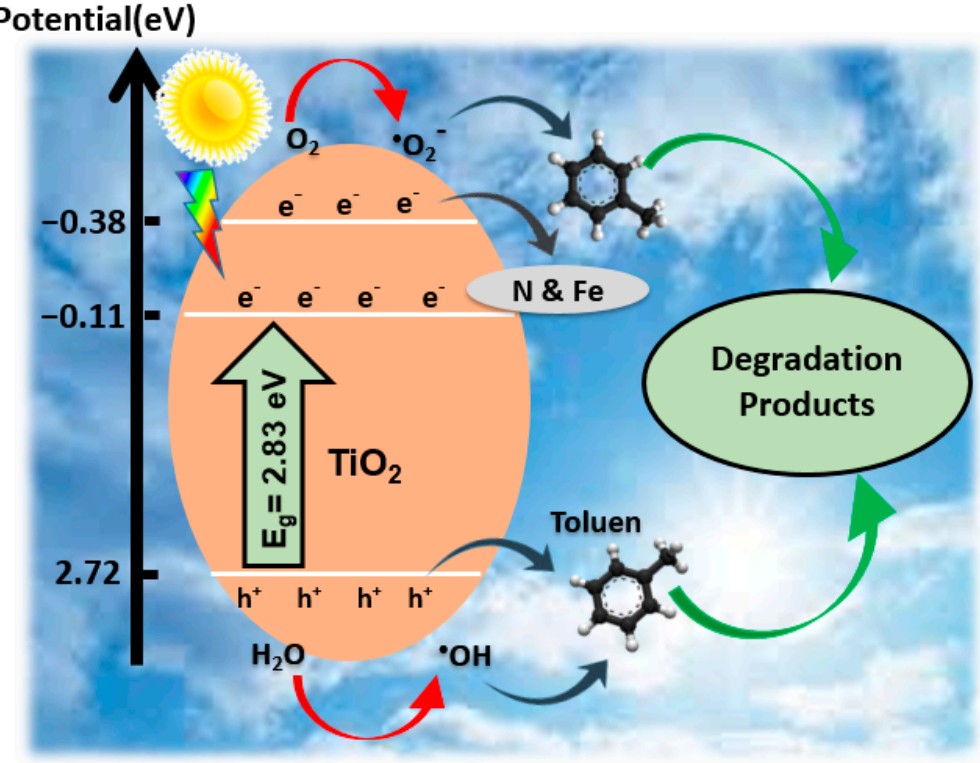

**Figure 14.** Reaction mechanism of photodegradation of toluene over FeNTiO$_2$ (5)-U nanophotocatalyst.

In this equation, X is the geometrical mean of the electronegativity of atoms, $E_0$ is the energy of free electrons in the hydrogen scale that is +4.5 eV, $E_g$ is the band gap energy, $E_{cb}$ is the energy of the conduction band and $E_{vb}$ is the energy of valance band.

Based on these equations, $E_{cb}$ and $E_{vb}$ of FeNTiO$_2$ (5)-U are $-0.11$ and $+2.72$ eV. Because $E_{vb}$ of this sample is more than that of the potential level of H$_2$O/●OH (2.7 eV), pores in the valance band can form OH radicals. Pores on the surface and OH radicals can react with pollutant molecules and degrade them. On the other hand, the reduction–oxidation potential of O$_2$/−●O$_{2-}$ is $-0.28$ eV, and conduction band electrons cannot produce peroxide radicals. The lowest wavelength energy in the current study is 3.93 eV, so electrons of the valance band can take this energy and excite it to upper levels. These electrons can produce superoxide from absorbed oxygen to the photocatalyst surface. Electrons with small energies (0.7 eV), which exist at low levels, can produce H$_2$O$_2$. Superoxide radicals and hydrogen peroxide can attack pollutant molecules and produce CO$_2$, H$_2$O, and other by-products. Specific surface area FeNTiO$_2$ (5)-U is large and improves absorption of pollutants on their surface. Finally, contact between pollutant and photocatalyst increases and promotes degradation rate. Reaction to degradation of toluene follows:

$$\text{FeNTiO}_2(5)\text{-U} + h\vartheta \rightarrow \text{FeNTiO}_2(5)\text{-U (e− + h}^+) \tag{1}$$

$$\text{FeNTiO}_2(5)\text{-U (h}^+) + \text{H}_2\text{O} \rightarrow \text{FeNTiO}_2(5)\text{-U} + \text{H+ }^\bullet\text{OH} \tag{2}$$

$$\text{FeNTiO}_2(5)\text{-U (h}^+) + \text{OH}^- \rightarrow \text{FeNTiO}_2(5)\text{-U} + {}^\bullet\text{OH} \tag{3}$$

$$\text{FeNTiO}_2\text{(5)-U (e-)} + \text{O}_2 \rightarrow \text{FeNTiO}_2\text{(5)-U} + {}^\bullet\text{O}_2- \tag{4}$$

$${}^\bullet\text{O}_2 + \text{H+} \rightarrow {}^\bullet\text{H}_2 \tag{5}$$

$$\text{HO}^\bullet{}_2 + \text{HO}^\bullet{}_2 \rightarrow \text{H}_2\text{O}_2 + \text{O}_2 \tag{6}$$

$$\text{FeNTiO}_2\text{(5)-U (e-)} + \text{H}_2\text{O}_2 \rightarrow {}^-\text{OH} + {}^\bullet\text{OH} \tag{7}$$

$$\text{Pollutant} + {}^\bullet\text{O}_2- + {}^\bullet\text{OH} \rightarrow \text{H}_2\text{O} + \text{CO}_2 + \text{Other products} \tag{8}$$

While this study presents practical insights, it comes with both advantages and limitations, much like other research in the field. One of the primary strengths of this study is the implementation of a real-based two-stage continuous slope reactor, which closely simulates actual workplace conditions. This setup not only enhances the retention time but also increases the interaction between the pollutant and the photocatalyst surface, thereby potentially improving the overall efficiency of the photocatalytic process. The use of an ultrasonic method to synthesize the photocatalyst is another important advantage, as it increases the photocatalyst's activity under visible light irradiation, making the procedure more effective and sustainable by employing a wider spectrum of light.

However, despite these strengths, the study also faces certain limitations. A notable drawback is the difficulty in synthesizing bulk volumes of the photocatalyst, primarily due to the requirement of additional starting materials, which can be cost-prohibitive and resource-intensive. Additionally, while the ultrasonic synthesis technique improves photocatalyst activity, it may concern complicated equipment and precise control situations that could pose challenges in replicating the process consistently at a larger scale.

These challenges emphasise areas for future research, such as investigating alternative synthesis techniques that could synthesise larger numbers of photocatalysts without significantly increasing costs or resource demands, and exploring methods to facilitate the ultrasonic synthesis method for easier scalability. By addressing these challenges, the potential for practical, large-scale application of the photocatalytic process can be further realized, contributing to more effective pollutant degradation in real-world settings.

## 4. Conclusions

In this study, Fe-N/TiO$_2$ photocatalysts were prepared using both ultrasonic-assisted co-precipitation and a hydrothermal method. The ultrasonic-assisted method resulted in smaller particle sizes and higher specific surface areas (S$_{\text{BET}}$), enhancing the photocatalytic activity of the synthesized samples. Among all the samples, Fe-N/TiO$_2$ (5) U demonstrated the highest photocatalytic performance for degrading gaseous toluene within 15 s. BET analysis revealed that the specific surface area of the sample synthesized with ultrasound was 88.3 m$^2$/g, compared to 74 m$^2$/g for the non-ultrasonic-assisted sample, highlighting the role of ultrasound in increasing the SBET of the photocatalysts. XRD analysis showed that the crystalline size of the non-ultrasonic sample was larger than that of the ultrasonic-assisted sample. SEM images indicated that the ultrasonic-assisted sample had significantly less agglomeration, which positively affected the toluene removal rate. Additionally, EDX and FTIR analyses confirmed the successful doping of Fe and N into the TiO$_2$ lattice. The ultrasonic-assisted photocatalyst also had a smaller band gap energy than the non-ultrasonic sample, contributing to its superior photocatalytic performance under visible light. The Fe-N/TiO$_2$ (5) U photocatalyst achieved a 63.5% degradation rate, while the non-ultrasonic-assisted sample achieved only 50%. This study confirms the significant synergistic effect of ultrasound and co-doping in enhancing photocatalytic activity for the degradation of gaseous toluene. Despite the progress made in the photocatalytic degradation of air pollutants, future research should build on the core concept of this study, the use of a continuous reactor, and focus on its application in work environments. Additionally, future studies like this one, which has employed ultrasonic methods and elements such as iron and nitrogen to enhance performance in the visible light spectrum, should explore techniques that further improve photocatalyst efficiency in this range. This is important because ultraviolet radiation has harmful health effects.

## 5. Patents

This article is derived from a master's thesis at Tabriz University of Medical Sciences. The reactor used in the study was patented in Iran's Real Estate and Deeds Registration Organization with registration number 108566 and international classification A61L9/20 on 16 January 2021.

**Author Contributions:** Conceptualization, O.A. and H.G.; Methodology, O.A., Y.R. and M.H.; Formal analysis, O.A., M.H. and A.T.; Investigation, O.A.; Writing—original draft, O.A.; Writing—review & editing, Y.R., M.H. and H.G.; Supervision, Y.R., M.H. and H.G. All authors have read and agreed to the published version of the manuscript.

**Funding:** This research was funded by Tabriz University of Medical Sciences.

**Data Availability Statement:** Data are contained within the article.

**Acknowledgments:** The authors extend their sincere gratitude to the Iran National Science Foundation, Tabriz University of Medical Sciences, and Sahand University of Technology for their financial support of this research.

**Conflicts of Interest:** The authors declare no conflict of interest.

## Abbreviations

| Abbreviation | Complete Term |
| --- | --- |
| PTFE | Poly Tetra Fluoro Ethylene |
| XRD | X-ray Diffraction |
| BET | Brunauer–Emmett–Teller |
| DRS | Diffuse Reflection Spectroscopy |
| FTIR | Fourier Transform Infrared Spectroscopy |
| EDX | Energy Dispersive X-ray Spectroscopy |
| SEM | Scanning Electron Microscope |
| VOCs | Volatile Organic Compounds |
| IRAC | International Agency of Research and Cancer |
| EPA | Environmental Protection Agency |
| NIOSH | National Institute for Occupational Safety and Health |
| OSHA | Occupational Safety and Health Administration |
| PPE | Personal Protective Equipment |
| UV | Ultraviolet irradiation |
| PCO | Photocatalytic oxidation |

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
