# Peer review of "Investigation of Synergistic Influence of Ultrasound and Co-Doping to Degrade Toluene from Polluted Air in Construction Sites—An Experimental Approach"

_buildings, doi:10.3390/buildings14092876_

Round 1

Reviewer 1 Report

Comments and Suggestions for Authors

Reviewer’s Report on the manuscript entitled:

Investigation of Synergistic Influence of Ultrasound and Co-doping to Degrade Toluene from Polluted Air in Construction Sites – An Experimental Approach

The authors synthesized photocatalysts based on pure and modified TiO2 to evaluate their efficacy in degrading gaseous toluene, a prevalent air pollutant in construction settings. In general, the topic and results are interesting, and the manuscript is well-written. I have some suggestions for further improvement.

Please highlight the main contributions of your work at the end of the Introduction, preferably using bullet points.

Figures 1 and 3. Please improve the resolution and font size of these two figures. Your other figures are good. Also please increase the font size of texts and numbers in the left panels of Figure 7.

Section 3. Please separate the Discussion from the results section. The result section should have all the results including figures and tables, while the discussion section should discuss and interpret the results in the light of similar studies. Please also discuss the advantages and limitations of your work in this section.

Section 3.3. The equations in this section belong to the method section not the results section.

Figure 15. Did you produce this figure or is it modified from somewhere else? Please ensure there is no copyright issue.

Acknowledgement. “he authors…” please replace it with “The authors…”
In the patent section you wrote: “January 26, 1401”. Please write the year in miladi not shamsi.

Thank you!

Comments on the Quality of English Language

There are some grammar/typo/style issues that should be checked and corrected.

Author Response

Comments 1: Please highlight the main contributions of your work at the end of the Introduction, preferably using bullet points.

Response: Thanks for the constructive comment.

The following are added to the end of introduction in page 4.  

Finally, main contributions to the body of knowledge and the practice can be listed as:

  • Set up a continuous air purification reactor suitable for the real conditions of working environments,
  • -Increasing the contact surface of the pollutant with the photocatalyst and the retention time by creating sloping surfaces,
  • -synthesis a photocatalytic with better properties using ultrasonic bath technique,
  • Achieving a high rate of toluene degradation under continuous conditions, and
  • Improvement of photocatalyst performance under visible light compared to ultraviolet light

Comments 2:Figures 1 and 3. Please improve the resolution and font size of these two figures. Your other figures are good. Also please increase the font size of texts and numbers in the left panels of Figure 7.

Response: Thank you for your feedback and constructive suggestion. the resolution and font size of these two figures are improved in the revised version.

Comments 3: Section 3. Please separate the Discussion from the results section. The result section should have all the results including figures and tables, while the discussion section should discuss and interpret the results in the light of similar studies. Please also discuss the advantages and limitations of your work in this section.

Response: Thank you for your valuable feedback. In this study, similar to several studies related to chemistry and laboratory research, where results are often presented as experimental data, separating the results from the discussion can reduce the readability of the paper. Please see below some examples of articles following this pattern, reflecting a common practice in this field. We would like to draw the attention of the respected reviewer to the Buildings journal's guidelines where it permits the combination of results and discussion sections. Therefore, the authors would like to keep the current form of presenting the results and discussion together.

-Rafiq A, Ikram M, Ali S, Niaz F, Khan M, Khan Q, Maqbool M. Photocatalytic degradation of dyes using semiconductor photocatalysts to clean industrial water pollution. Journal of Industrial and Engineering Chemistry. 2021 May 25;97:111-28.

-Eftekharipour F, Jamshidi M, Ghamarpoor R. Fabricating core-shell of silane modified nano ZnO; Effects on photocatalytic degradation of benzene in air using acrylic nanocomposite. Alexandria Engineering Journal. 2023 May 1;70:273-88.

-Xu, Jun, Haijun Lu, Zhenhua Wang, Qian Zhang, Guanghua Cai, and Meng Zang. 2024. "Experimental Study on Transport of Cd(II) and Cu(II) in Landfill Improved Clay Liners Building Material Containing Municipal Sludge-Activated Carbon" Buildings 14, no. 9: 2638. https://doi.org/10.3390/buildings14092638

As for the discussion on the advantages and limitations of this study, the followings are added to page 21, at the end of results and discussion:

While this study presents practical insights, it comes with both advantages and limitations, much like other research in the field. One of the primary strengths of this study is the implementation of a real-based two-stage continuous slope reactor, which closely simulates actual workplace conditions. This setup not only enhances the retention time but also increases the interaction between the pollutant and the photocatalyst surface, thereby potentially improving the overall efficiency of the photocatalytic process. Also, the use of an ultrasonic method to synthesize the photocatalyst is another important advantage, as it increases the photocatalyst's activity under visible light irradiation, making the procedure more effective and sustainable by employing a wider spectrum of light.

However, despite these strengths, the study also faces certain limitations. A notable drawback is the difficulty in synthesizing bulk volumes of the photocatalyst, primarily due to the requirement of additional starting materials, which can be cost-prohibitive and resource-intensive. Additionally, while the ultrasonic synthesis technique improves photocatalyst activity, it may concern complicated equipment and precise control situations that could pose challenges in replicating the process consistently at a larger scale.  

These challenges emphasise areas for future research, such as investigating alternative synthesis techniques that could synthesise larger amounts of photocatalysts without significantly increasing costs or resource demands and exploring methods to facilitate the ultrasonic synthesis method for easier scalability. By addressing these challenges, the potential for practical, large-scale application of the photocatalytic process can be further realized, contributing to more effective pollutant degradation in real-world settings.

Comments 4: Section 3.3. The equations in this section belong to the method section not the results section.

Response: Thanks for your comment. This section is typically presented in several similar studies within the results and discussion part. Essentially, it’s important to discuss the reaction pathway of toluene with the photocatalyst under light irradiation and its resulting products. Below are some examples for your reference.

-Ma D, Yang L, Sheng Z, Chen Y. Photocatalytic degradation mechanism of benzene over ZnWO4: Revealing the synergistic effects of Na-doping and oxygen vacancies. Chemical Engineering Journal. 2021 Feb 1;405:126538.

-Rao Z, Shi G, Wang Z, Mahmood A, Xie X, Sun J. Photocatalytic degradation of gaseous VOCs over Tm3+-TiO2: Revealing the activity enhancement mechanism and different reaction paths. Chemical Engineering Journal. 2020 Sep 1;395:125078.

Comments 5: Figure 15. Did you produce this figure or is it modified from somewhere else? Please ensure there is no copyright issue.

Response: Thanks for your comment.

All the text and chemical reaction pathways in the image were created by the authors via using PowerPoint software. In addition, the authors can confirm there is no copyright issue with any of the figures.

Comments 6: Acknowledgement. “he authors…” please replace it with “The authors…”
In the patent section you wrote: “January 26, 1401”. Please write the year in miladi not shamsi.

Response: Thank you for your comment. The errors you mentioned have been corrected. These modifications, “January 16, 2021” and “The authors extend”. have been implemented and highlighted in the Acknowledgement section.

Comments 7: Comments on the Quality of English Language. There are some grammar/typo/style issues that should be checked and corrected.

Response:  Thank you for your comment. A professional proofreading is conducted to address any language issues.

Reviewer 2 Report

Comments and Suggestions for Authors

1. The abstract is long. I felt that the background content in the abstract is not necessary. You may write the abstract with methods and contributions along with the results of your work.

2.  Please avoid the lumped references in the introduction section. Each article must be carefully examined, and its technical merits and drawbacks should be included. 

3. The author's contribution must be highlighted with bullet points at the end of the introduction section. 

4. Please create a "nomenclature" to indicate the variables and abbreviation before starting the introduction section. 

5. The inference from Fig. 5 is not clear. Kindly update the content. 

6. What is the future direction or extension of this work. It must be highlighted in the conclusion section. 

Author Response

Comments 1: The abstract is long. I felt that the background content in the abstract is not necessary. You may write the abstract with methods and contributions along with the results of your work.

Response: Thank you for your helpful comments. The abstract of the article, particularly the background section, has been revised as much as possible, and any overwriting has been avoided.

Comments 2: Please avoid the lumped references in the introduction section. Each article must be carefully examined, and its technical merits and drawbacks should be included. 

Response: Thank you for your useful feedback about the use of lumped references in the introduction section. In some cases, however, it is necessary to refer to a group of studies to provide context and explain the background effectively. When doing so, we have aimed to summarize the key results and contributions of these studies while ensuring that each work is acknowledged properly and its relevance to the topic is clear. To address your concern, we checked the introduction section and made some adjustments to ensure that each reference is adequately discussed.

Comments 3: The author's contribution must be highlighted with bullet points at the end of the introduction section. 

Response: 

Thank you for your valuable feedback. Please see the amendments to the pages 3-4.

Finally, the main contributions to the body of knowledge and the practice can be listed as:

  • Set up a continuous air purification reactor suitable for the real conditions of working environments,
  • Increasing the contact surface of the pollutant with the photocatalyst and the retention time by creating sloping surfaces,
  • Synthesis a photocatalytic with better properties using ultrasonic bath technique,
  • Achieving a high rate of toluene degradation under continuous conditions, and
  • Improvement of photocatalyst performance under visible light compared to ultraviolet light.

Comments 4: Please create a "nomenclature" to indicate the variables and abbreviation before starting the introduction section. 

Response: 

Thank you for your input.

The list of terms and abbreviations has now been included before of the introduction section.

No

Acronym

Complete term(s)

1

PTFE

Poly Tetra Fluoro Ethylene

2

XRD

X-Ray Diffraction

3

BET

Brunauer-Emmett-Teller

4

DRS

Diffuse Reflection Spectroscopy

5

FTIR

Fourier Transform Infrared Spectroscopy

6

EDX

Energy Dispersive X-ray Spectroscopy

7

SEM

Scanning Electron Microscope

8

VOCs

Volatile Organic Compounds

9

IRAC

International Agency of Research and Cancer

10

EPA

Environmental Protection Agency

11

NIOSH

National Institute for Occupational Safety and Health

12

OSHA

Occupational Safety and Health Administration

13

PPE

Personal Protective Equipment

14

UV

Ultra-Violet irradiation

15

PCO

Photocatalytic oxidation

16

MFC

Mass Flow Controller

17

JCPDSs

Joint Committee on Powder Diffraction Standards

Comments 5: The inference from Fig. 5 is not clear. Kindly update the content. 

Response:  Thank you for your feedback; the content has been updated and highlighted in page 9.

The incorporation of iron and nitrogen into the TiO2 structure reduces both particle size and agglomeration. Figures 5c and 5d illustrate the surfaces of Fe-N/TiO2 (5) and Fe-N/TiO2 (5)-U, respectively. It is evident that Fe-N/TiO2 (5)-U particles are smaller and show no signs of agglomeration, indicating that the ultrasonic method enhances the distribution of Fe and N on the TiO2 surface.

Comments 6: What is the future direction or extension of this work? It must be highlighted in the conclusion section. 

Response:  Thank you for your constructive and helpful comment. The suggested changes have been incorporated into the conclusion and highlighted. Please see page 21.

"Despite the progress made in photocatalytic degradation of air pollutants, future research should build on the core concept of this study, the use of a continuous reactor, and focus on its application in work environments. Additionally, future studies, like this one, which employed ultrasonic methods and elements such as iron and nitrogen to enhance performance in the visible light spectrum, should explore techniques that further improve photocatalyst efficiency in this range. This is important because ultraviolet radiation has harmful health effects.”

Round 2

Reviewer 2 Report

Comments and Suggestions for Authors

Thank you for your revision. The response letter is fine. The article can be accepted in its current form.